# Umbilical Cord Prolapse—Interesting CTG Traces

**DOI:** 10.3390/diagnostics12112845

**Published:** 2022-11-17

**Authors:** Radu Botezatu, Nicolae Gica, Gheorghe Peltecu, Anca Maria Panaitescu

**Affiliations:** 1Department of Obstetrics and Gynecology, Filantropia Clinical Hospital, 71117 Bucharest, Romania; 2Department of Obstetrics and Gynecology, Carol Davila University of Medicine and Pharmacy, 71117 Bucharest, Romania

**Keywords:** CTG, cord prolapse, acute hypoxia, obstetrical emergency

## Abstract

Umbilical cord prolapse can be a life-threatening obstetrical event involving the fetus due to sudden oxygenated blood flow obstruction. These types of events most often happen in labor and are associated with obstetric maneuvers. Rarely, a clinical examination can diagnose the condition, but the situation is usually detected secondary to an abnormal cardiotocography trace. We present several clinical cases where a CTG trace was used to infer umbilical cord prolapse.

## 1. Introduction

Umbilical cord prolapse is a rare clinical situation with a reported incidence of 1 to 6 per 1000 pregnancies, but decreasing lately, presumably due to the reduced number of grand multiparous women and the increasing incidence of cesarean section [1]. The situation is described when the cord slips through the dilated cervix below the presenting part of the fetus. In most cases, prolapse is a clinical emergency given the fact that the cord is vulnerable to compression between the presenting part and maternal soft tissues (especially the cervix). 

In 2021 Wong et al. [2] revised the definition and management of umbilical cord prolapse and classified the situation into three distinct categories: cord prolapse (when the cord is below the cervical os), cord presentation (when the cord is above the cervix but below the presenting part), and compound cord presentation (when the cord is above the cervix and lateral to the presenting part). 

There are several well-known risk factors associated with cord prolapse, such as obstetrical interventions (artificial rupture of membranes (ARM), if the presenting part is mobile or high [3]), external cephalic version with spontaneous rupture of membranes, and induction of labor with transcervical balloon catheters [4]. Other risk factors include fetal malpresentation, unstable lie, preterm labor, or polyhydramnios [5]. 

When the clinical situation is diagnosed late or misdiagnosed, a very high degree of perinatal mortality is associated [3]. 

Given the fact that it is a relatively rare event, specific training for correct/appropriate management is often required in all obstetric services. Clinical simulation plays an important role in preventing associated complications [6,7]. 

When umbilical cord prolapse is diagnosed, emergency delivery is indicated to prevent fetal demise. Leong et al. also described conservative management in extremely preterm pregnancies, but this is considered an exceptional situation [8]. 

The decision-to-incision interval has also been studied. Wasswa et al. noted that an interval of less than 30 min is associated with better outcomes [9]. Tashfeen et al. also noted that lower Apgar scores (less than 7 at 5 min) are associated with a decision-to-incision interval longer than 60 min (58.3% vs. 3.2%) [10]. Several other small studies reported a long optimal decision-to-incision interval time, mostly because of a high variation in associated clinical settings (place of diagnosis, availability of operating theatre, clinical maneuvers applied before delivery to prevent blood flow obstruction) [11,12,13]. 

The decision-to-incision parameter is not always the best predictor for perinatal outcome as it depends on the moment of diagnosis. When there is a late diagnosis with a compromised fetus, the decision-to-incision time interval is less relevant. 

However, in order to maintain the best neonatal outcome associated with this clinical issue, a more accurate method of diagnosis is required. Specific cardiotocographic (CTG) changes have been associated with suspected diagnosis, and therefore any pathological trace which is associated with known risk factors must raise suspicion of cord prolapse. 

When a decision to delivery/deliver is made, apart from reducing decision-to-incision time, several actions need to be taken to prevent acute obstruction during preparation intervals. 

Several maneuvers have been described to prevent complete obstruction, such as manual or indirect disengagement of the presentation/presenting part and specific maternal positions to elevate the maternal pelvis in order to use gravitation for pulling up the presenting part. 

Wong et al. [2] described clinical maneuvers that can be applied during preparation for delivery. These consist of transvaginal manual elevation (pushing up the fetal presenting part in order to decompress the umbilical cord), filling the maternal urinary bladder with saline (with the effect of pushing up the presenting part indirectly), using gravity positions (the knee-chest position, Trendelenburg position), or wedging the maternal pelvis in the supine and lateral positions, respectively. 

## 2. CTG Traces in Umbilical Cord Prolapse

When analyzing CTG traces, Wong et al. showed that cord prolapse may be associated with persistent bradycardia, recurrent decelerations, or a normal trace [14]. In the same study, the authors mentioned that the pH of arterial blood decreased by 0.009 per minute with a high risk of acidosis when delivery was accomplished after 20 min. 

Persistent bradycardia as a continuation of a prolonged deceleration is one of the main irreversible causes of intrapartum acute hypoxia that requires expedited delivery [15]. 

In the acute hypoxia pattern, the umbilical artery pH drops by 0.1 every 10 min. In this situation, the 3–6–9–12–15-min rule must be applied (intrauterine resuscitation by 6 min, move the patient to a suitable delivery location by 9 min, if CTG trace is persistently bradycardic, commence delivery procedures by 12 min and deliver the baby by 15 min) [16]. 

**Figure 1 diagnostics-12-02845-f001:**
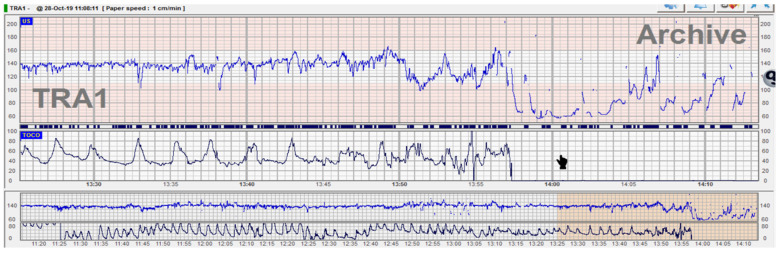
Umbilical cord prolapse associated with acute bradycardia. We present a case of a para 3 patient who was admitted in active labor at 8 cm dilation with intact membranes. On vaginal examination, the cervix was fully effaced with bulging membranes, and therefore she was directed to a labor room. CTG trace was normal, and therefore she was proposed for controlled ARM (artificial rupture of membranes) which she declined. During monitoring, spontaneous rupture of membranes occurred and the fetal heart rate (FHR) trace suddenly dropped to 60–70 bpm. On emergency vaginal examination, the umbilical cord loop was found beneath the fetal head and between the cervix and the fetal skull. Because the patient was in active labor, it was impossible to mobilize the fetal head upwards. Emergency cesarean section was performed, and a healthy baby was delivered within 15 min of diagnosis with Apgar scores 6 and 9 at 1 and 5 min. When analyzing the trace, we can classify this CTG trace as normal (stable baseline, normal variability, and no signs of evolving hypoxia) therefore we can theoretically estimate a normal cord blood pH (for example 7.2). When the acute event happened, a diagnosis was made/established in less than 3 min. With an acute hypoxia pattern (prolonged deceleration) the pH drops by 0.1 every 10 min, and therefore, in less than 20 min the pH will still be above 7, which is considered normal.

**Figure 2 diagnostics-12-02845-f002:**
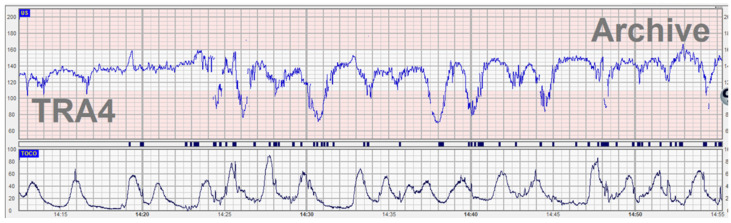
Abnormal CTG (recurrent decelerations) associated with compound cord presentation. A G2 P2 patient with ruptured membranes one hour before being admitted to the labor ward in the early stage due to an intermediary CTG trace—recurrent variable decelerations chemoreceptor-type. She had 3–4 unpainful irregular uterine contractions in 10 min. On vaginal examination, the cervix was 2 cm dilated. Lateral to the fetal head, an umbilical loop was palpated (a compound cord presentation). An emergency cesarean section was performed with a healthy baby delivered in normal conditions. After delivery, when analyzing the CTG trace we found an abnormal clinical association with the subacute hypoxia model which is often seen in the active first stage of labor. At 2 cm dilation, the intermediary trace is usually associated with subacute pathology (temporary umbilical cord obstruction), but in our presented case the fetus started to activate a compensation mechanism, releasing catecholamines with tachycardia, and therefore a clinical examination of the cervix was performed.

**Figure 3 diagnostics-12-02845-f003:**
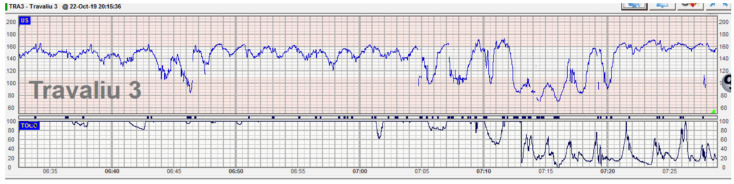
Prolonged deceleration associated with umbilical cord prolapse. This case is about a primiparous patient in the latent first stage of labor. She was admitted for regular contractions at 36 weeks with rupture of membranes, cephalic presentation at −1 station, and the cervix 5 cm dilated. CTG trace was classified as normal with a stable baseline and reassuring variability with decelerations. Fetal heart rate in the upper segment of the normal range was considered normal given the gestational age. Suddenly, the patient reported the loss of a large amount of clear amniotic fluid. Immediately after, a prolonged deceleration was noted on the continuous CTG trace, the patient was examined, and a loop of the umbilical cord was detected in the upper third of the vagina. Tocolysis (hexoprenaline 10 mcg iv) was administered, and a member of the team was holding the presenting part to release cord compression while preparations for emergency cesarean section were in progress. An emergency cesarean section was performed, and the fetus was delivered with Apgar scores of 7 and 8 at 1 and 5 min.

## 3. Discussions

In all presented cases the umbilical cord prolapse was diagnosed based on the sudden changes in the CTG traces. However, all patients had risk factors for this pathology which were spotted during the initial clinical and anamnestic assessment. 

Cord prolapse is an obstetrical emergency where an expedited delivery is needed to minimize the hypoxic-ischemic risk to neonatal brain tissue. 

The risk is associated with complete or partial umbilical blood flow obstruction which corresponds with specific changes on the CTG trace. The complete obstruction of the umbilical cord is relatively rare (0.14–0.62%) [17]) and therefore the risk of brain injury is also low. However, given the fact that it is a life-threatening clinical condition, a high level of suspicion is needed when the risk factors are associated with intermediary CTG traces. 

Healthcare practitioners from the labor ward department should also distinguish between other causes of abnormal CTG traces and prepare for immediate action. An acute hypoxia pattern (prolonged deceleration or bradycardia) should expedite the delivery in irreversible cases such as uterine rupture and abruption and immediate treatment or intrauterine resuscitation is needed in hypertonia or epidural-induced maternal hypotension. In all other cases, the 3–6–9–12–15 min rule should be applied [18], but only after acute accidents have been excluded. 

## 4. Conclusions

Umbilical cord prolapse is one of the irreversible obstetrical emergencies associated either with active labor or prelabor and the third trimester of pregnancy. The gold standard of diagnosis is vaginal clinical examination after a suspected clinical or paraclinical event. When the event happens during hospitalization, the outcome is usually favorable due to rapid intervention in a suitable clinical environment. However, when the event happens in outpatient locations the risk of fetal demise is very high. Therefore, high-risk patients need to be antenatally counseled about cord prolapse and what actions could be undertaken before admission to the hospital to prevent brain damage. 

When a persistent deceleration is recorded on the CTG trace in labor, after spontaneous or ARM, cord prolapse has to be taken into consideration and urgently checked for.

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
