# Peer review of "Umbilical Cord Prolapse—Interesting CTG Traces"

_diagnostics, 2022, doi:10.3390/diagnostics12112845_

Round 1

Reviewer 1 Report

This is certainly an interesting comment.  My only issue with the paper is that the figures (1 and 3 especially) do not reflect what they are stated to reflect.  For example, the first case is "acute bradycardia".  The fetal tracing in the paper does not show a bradycardia (generally defined as a HR < 110 BPM for 10 mins or longer).  The tracing is not able to be interpreted at the end since the signal is lost.  The third case is supposed to show a "prolonged deceleration".  The selected tracing does not show a prolonged deceleration.  We do not know what the tracing shows because it ends as the tracing changes.

Minor comment:  line 27 - cervical "os" not ox

Author Response

This is certainly an interesting comment.  My only issue with the paper is that the figures (1 and 3 especially) do not reflect what they are stated to reflect.  For example, the first case is "acute bradycardia".  The fetal tracing in the paper does not show a bradycardia (generally defined as a HR < 110 BPM for 10 mins or longer).  The tracing is not able to be interpreted at the end since the signal is lost.  The third case is supposed to show a "prolonged deceleration".  The selected tracing does not show a prolonged deceleration.  We do not know what the tracing shows because it ends as the tracing changes.

      Response: Giving the fact that we do not have any other images from those cases to clearly reflect prolonged deceleration and bradycardia, we replaced first and third case with new ones and also replaced ctg traces.

Minor comment:  line 27 - cervical "os" not ox

Response: Replace was done. thank you

Reviewer 2 Report

This short text presents three cases of umbilical cord prolapse with respective CTG traces. I have the following comments:

1.       The introduction section should be shortened, each paragraph needs proper referencing and important international guidelines (RCOG, ACOG….) should be mentioned.

2.       How many cases of umbilical cord prolapse are diagnosed in the authors’ institution each year?

3.       On the first image the bradycardia is not clearly visible

4.       In the discussion section the authors should briefly discuss other possible causes of abnormal CTG traces

Author Response

This short text presents three cases of umbilical cord prolapse with respective CTG traces. I have the following comments:

  1. The introduction section should be shortened, each paragraph needs proper referencing and important international guidelines (RCOG, ACOG….) should be mentioned.

Introduction was shortened and also reference to guidelines were added

  1. How many cases of umbilical cord prolapse are diagnosed in the authors’ institution each year?

We diagnose aprox 5 cases per year from 2500 vaginal deliveries.

  1. On the first image the bradycardia is not clearly visible

Giving the fact that we do not have any other images from those cases to clearly reflect prolonged deceleration and bradycardia, we replaced first and third case with new ones and also replaced ctg traces.

  1. In the discussion section the authors should briefly discuss other possible causes of abnormal CTG traces

We added discussion about other possible cases of abnormal ctg (reversible and irreversible situations).

Round 2

Reviewer 1 Report

Comments addressed.  Minor comments:

Figure 1 legend:  

- line 95 (8 or 4 cm?)

- line 104 - delete "Category 1"

Figure 3 legend:

- Line 129:  delete "with presenting part" ?

- Line 133:  delete "uncomplicated".  "Uncomplicated deceleration" is not a term that is used.  

- Line 135:  delete "with parasympathetic nervous system immaturity".  

- Line 138:  What kind of tocolysis was administered in this emergency situation.  I would either delete it or be specific regarding the medication administered.

- Line 140:  delete "in"

- Page 4:  Line 157:  earlier referred to as 3-6-9-12-15 rule.  Include 15 here for consistency.

Author Response

Dear reviewers,

Thank you again very much for your time and your valuable indications. We revised manuscript according to your comments.

Reviewer 1

Comments addressed.  Minor comments:

Figure 1 legend:  

- line 95 (8 or 4 cm?)

- line 104 - delete "Category 1"

Figure 3 legend:

- Line 129:  delete "with presenting part" ?

- Line 133:  delete "uncomplicated".  "Uncomplicated deceleration" is not a term that is used.  

- Line 135:  delete "with parasympathetic nervous system immaturity".  

- Line 138:  What kind of tocolysis was administered in this emergency situation.  I would either delete it or be specific regarding the medication administered.

- Line 140:  delete "in"

- Page 4:  Line 157:  earlier referred to as 3-6-9-12-15 rule.  Include 15 here for consistency.

All changes were done

Reviewer 2 Report

All comments have been addressed and I believe the manuscript is now suitable for publication

Author Response

Thank you.